

# SH-SDS: a new static-dynamic strategy for substation host security detection

Yang Diao[1], Hui Chen[2], Wei Liu[1] and Abdur Rasool[3]

[1] Shaoguan Power Supply Bureau, Guangdong Power Grid Co., Ltd., Shaoguan, Guangdong, China
[2] School of Artificial Intelligence, Shenzhen Polytechnic University, Shenzhen, GuangDong, China
[3] Shenzhen Key Laboratory for High Performance Data Mining, Shenzhen Institute of Advanced Technology, Chinese Academy of Sciences, Shenzhen, Guangdong, China

## ABSTRACT

A substation is integral to the functioning of a power grid, enabling the efficient and safe transmission and distribution of electrical energy to meet the demands of consumers. The digital transformation of critical infrastructures, particularly in the electric power sector, such as the emergence of intelligent substations, is a double-edged sword. While it brings about efficiency improvements and consumer-centric advancements, it raises concerns about the heightened vulnerability to cyberattacks. This article proposes a new static-dynamic strategy for host security detection by implementing a system prototype and evaluating its detection accuracy. To reduce the subjectivity in manually selecting features, we combine classified protection for cybersecurity-related standards and construct the requirement generation algorithm to construct a network security detection standard library for the substation host. Based on this, we develop strategy generation algorithm to match the list of host detection projects to obtain the security detection strategy of the target host. Moreover, we output and analyze the detection logs to obtain a security detection report. The prototype is efficient and effective through practical use, and it serves as a practical tool in substation host security detection. The experiments suggest that the mechanism proposed in our study can operate at a high speed and demonstrates satisfactory performance in terms of detection.

## INTRODUCTION

A power grid, or an electrical or energy grid, is a complex network of interconnected components designed to generate, transmit, and distribute electrical power. It is a crucial infrastructure that forms the backbone of modern society, providing electricity to homes, businesses, industries, and various institutions. The substation is a critical component in an electrical power system, serving as an intermediary point between the high-voltage transmission system and the low-voltage distribution system.

Previously, the grid information could not be efficiently shared in traditional power grids due to the relative independence of information platforms like relay protection management information system, environment supervision system, and SCADA system (Supervisory Control and Data Acquisition System) (*Hegde et al., 2021*). However, with the

Corresponding author
Hui Chen, hui.chen1@siat.ac.cn

development of network communication, cloud computing, big data, and other technologies, data interaction between different power grid platforms has become possible. The digitization of the power grid is essential to promoting the country's construction of information power systems.

The digital power grid is a new energy ecosystem driven by a core of next-generation digital technology, with data including the collection, analysis, and utilization as a critical production factor, and built upon modern power energy networks and a new generation of information networks. As an essential node in the power grid, the digital transformation of substations is fundamental. However, the application of Internet technology in power grids, such as microprocessor-based IEDs (intelligent electronic devices), increases the risk of substation network security, which is mentioned as a source of previously unknown security problems, especially in digital substations (*Huang et al., 2023*). These substations are distributed throughout the country in various unsupervised locations, making them potential targets for cyber threats. If the substation host is invaded, it may lead to unplanned power outages or other power accidents, affecting the reliability of the power supply. In 2016, a breach in the power system domain accounted for 20% of reported cyberattack occurrences. Hackers are creating new cyberattack techniques with the power grid in mind, such as exploiting weaknesses in power sector protocols like TCP/IP, Ethernet, and connections to WANs by Internet technologies (*Zhang & Li, 2023*; *Li & Lei, 2019*; *Kim et al., 2019*). All of these emphasize the importance of robust cybersecurity measures in ensuring the reliability and security of the power grid.

In substations, embedded devices, and hosts are connected through the local area network and to the dispatching data network only through the dedicated vertical encryption authentication gateway. Network security attacks on substations can only be carried out by physical intrusion to the host on the spot or network intrusion through the dispatching data network. The vertical encryption authentication gateway adopts certificate-based authentication and data encryption transmission technology based on tunnel and power-specific encryption algorithms. It plays an excellent protective role against attacks from the dispatching data network. The hosts' strategy configuration can limit the physical intrusion from the spot. Thus, this article focuses on the detection of hosts' strategy configuration.

Currently, the network security detection of the substation host is carried out manually. The engineer inputs commands item by item according to the inspection file to check the reinforcement projects of the host network security. However, there are some shortcomings in the way manual inspection is performed. Firstly, manual inspection is slow and inefficient. It takes about 30 min to perform security checks on a host. Secondly, it isn't easy to guarantee the accuracy of the results in manual inspection. Finally, when the network security requirements for hosts are updated, it takes a lot of time to manually conduct security checks again on all hosts according to the new requirements. To address these challenges, this article introduces an innovative static-dynamic strategy. By automating feature selection and integrating a classified protection framework, we craft a standard library for network security detection. We develop RG (requirement generation) and SG (strategy generation) algorithms and streamline the alignment of detection projects

with tailored host security strategies. This strategy significantly enhances efficiency and accuracy in substation host security detection, as evidenced by our prototype's performance in practical applications and experiments, demonstrating high-speed operation and satisfactory results. This study offers two goals for substation host detection in practical application with the following features.

1) To improve the hosts' strategy configuration by setting up procedures to protect the hosts' operation, control the substation, and respond to different scenarios from unauthorized access and potential cyber threats.

2) To make a prototype that completes and updates network security checks and output reports automatically and efficiently.

In this scenario, the significant contributions of this article can be summarized as follows.

- A network security detection standard library is constructed for the substation host, and collects all items that need to be tested according to the network security testing requirements.
- We construct static detection libraries and dynamic detection libraries based on different types of projects and corresponding detection methods. A new SG algorithm is designed to match the list of host detection projects to obtain the security detection strategy of the target host.
- In practical applications, we develop the substation host security detection software, which has proven both efficient and effective, serving as a valuable tool in substation host security detection.

The rest of the article is organized as follows. 'Literature Review' briefly presents the related studies. 'Proposed Method: SH-SDS' discusses the grid modeling and the mathematical formulation of the controllers and discusses the case studies, and 'Experiments and Results' examines the scheme's stability and the effects of uncertainties. Finally, the 'Conclusion' concludes the article.

## LITERATURE REVIEW

### Power grid and substation

The power grid has evolved into a cyber-physical system, integrating intelligent communication and control technologies, which typically consists of power plants, electricity facilities (such as household electricity, industrial electricity, *etc.*), transmission towers, substations, and others, is used to transmit and distribute electric power (*Fang et al., 2012*), as shown in Fig. 1. For example, televisions and computers, as well as larger industrial machinery and equipment used in factories, are all electrical facilities integral to daily life and industrial processes, forming an essential part of the overall power system. Transmission towers serve as the support structures for transmission lines, playing a crucial role in fixing and supporting these lines to ensure the stability and reliability of the power system. Similarly, substations act as vital components within the power grid,

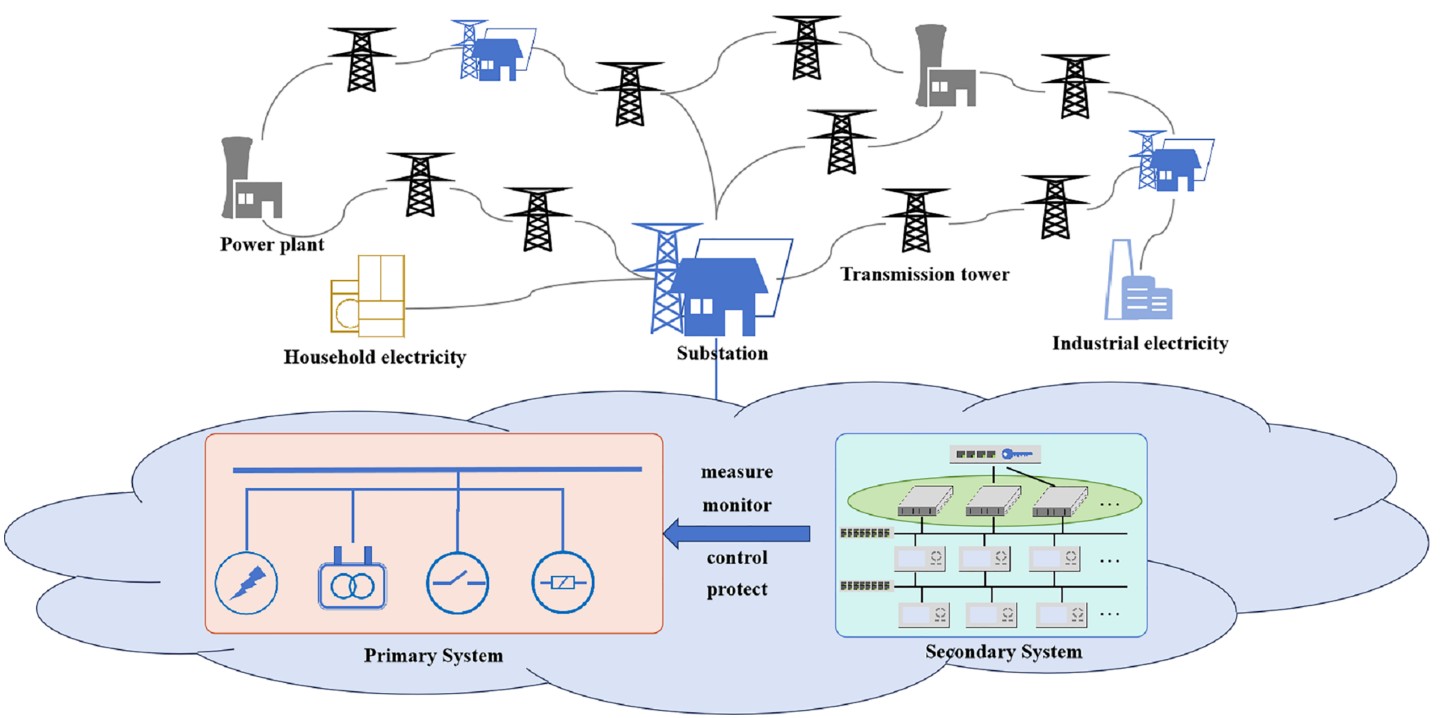

**Figure 1 Architecture of the modern grid including power plants, electricity facilities, transmission towers, substations and others.**

functioning as intricate nodes where voltage is transformed and regulated. The smart grid's cyber and physical components are deeply interconnected. Advances in communication and information technologies have been utilized to enhance the monitoring and control of power system. The interconnected nature of digital communications, remote control of devices, and data streams among intelligent machines create a larger attack surface. An attacker can potentially compromise cyber infrastructures and manipulate the control system, leading to physical damage.

Overall, the power grid is a sophisticated network that generates electrical power at power plants. It is converted and distributed through electricity facilities, and supported and controlled by transmission towers and substations. In modern power grids, integrating technology brings both advancements and challenges. Protecting against cyber threats becomes critical for the digital infrastructure and preventing potential physical damage to the power system.

The smart substation adopts advanced and reliable intelligent equipment, enhancing the information sharing and interoperability among intelligent power equipment and achieving the sharing of digital substation information, network communication platform, and standardized information (*Yuan et al., 2021*; *Huang, 2018*). The application of the standard IEC 68150 communication protocol in the smart substation provides a data platform for intelligent applications (*Kim et al., 2019*). Compared to traditional substations, intelligent substations have enhanced interoperability, reconfigurability,

controllability, maintainability, and flexibility (*Kezunovic et al., 2010*). However, with the improvement of substation intelligence capabilities, smart substations are also more susceptible to network attacks (*Tong et al., 2018*; *Yang et al., 2021*). In the past, private protocols have been commonly used for communication between different devices in substations, and the goal of ensuring network security is achieved by limiting the propagation range of communication protocols. However, the IEC 61850 and communication protocols used in intelligent substations are public. Network attacks on substations can more easily issue incorrect instructions to smart devices inside the substation by tampering with communication, leading to misoperation of devices inside the substation and causing negative impacts on the safe and stable operation of the power grid (*Gaspar et al., 2023*; *Karantaev et al., 2021*). Therefore, it is necessary to study the network security of intelligent substations.

## Cyberattack detection system in substations

The network security protection measures for smart substations can be divided into two methods: baseline verification and intrusion detection. The baseline verification improves the network security protection level of the host before the attack occurs (*Pattanavichai, 2017*), while intrusion detection methods detect and block attacks when it occurs (*AydN, Zaim & Ceylan, 2009*).

Security baseline verification is a passive security protection technology that includes system status, system security configuration, and system vulnerabilities (*Zhang & Li, 2023*; *Fang et al., 2012*; *Liu & Hu, 2023*; *Chen et al., 2017, 2021*; *Wang, Yang & Liu, 2020*; *Sun, 2020*; *Rotella, 2018*). It is the minimum requirement to ensure information security. By checking the baseline and rectifying issues, the overall security level of the host can be improved, achieving protection against attacks. For example, *Lina & Dongzhao (2020)* present a security SDS architecture which provided an open interface to change the network security configuration file, then resist network security attacks. *Zhang & Li (2023)* analyze the current security status of computer network information to propose an computer network information security protection evaluation method, which has an important practical significance for the security protection of enterprise computer network information. *Liu & Hu (2023)* design a security baseline automatic verification method based on the SCAP protocol, which takes into account the national standard for host security level protection, to achieve an improvement in the ability of domestic autonomous computer terminals to resist security risks. *Chen et al. (2021)* design a security baseline evaluation model that includes operational processes and scoring algorithms to suite for power intelligent IoT terminals. However, the main restriction of the existing research methods on baseline verification is that they hardly consider the network security of the substation secondary system.

IDSs (Intrusion Detection Systems) are security mechanisms designed to identify and respond to unauthorized activities or potential security incidents within a computer network or systems, which detect attacks when they happen or after they have taken place (*AydN, Zaim & Ceylan, 2009*). This distinction leads to two main types of IDSs: the signature-based IDSs and the anomaly-based IDSs. In real time, the signature-based IDSs

identify known attack patterns or signatures in network traffic or system logs, alerting when a match is found (*Jin, Chung & Xu, 2021*). The anomaly-based IDSs establish a typical network or system behavior baseline and trigger alerts or alarms when deviations from this baseline are detected. Anomalies may indicate potential attacks or security incidents (*Emanet, Karatas Baydogmus & Demir, 2023*). IDS plays a crucial role in enhancing the security posture of computer networks and systems by providing a proactive means to identify and respond to security threats. These are integral to a comprehensive cybersecurity strategy, working alongside other security measures such as firewalls, antivirus software, and security policies. *Sun (2020)* propose reinforcement methods for Windows hosts and constructed a multi-level active defense system to address security threats within enterprise networks. Table 1 provides insights into recent researches on other IDSs, including models or systems, used methods, and solving problems. However, most researches focus on methods to resist intrusion, and these are rarely used in substation secondary systems. Unlike ordinary intrusion detection, substation host security detection aims to connect through a local area network. It faces a relatively single type of intrusion and requires higher work efficiency when conducting intrusion detection.

In response to attacks that maliciously modify data, *Li et al. (2022)* enhance IEC61850 message security using a message authentication code function and dynamic keys in Hash-MAC calculations to minimize key cracking risks, alongside difference sequence variance for detecting abnormal packets, increasing detection accuracy. *Sun et al. (2022)* enhance the ProbeSpace self-attention and self-distillation mechanisms with Informer, addressing long sequence network traffic issues by focusing on time series features, thus boosting efficiency. *Valenzuela, Wang & Bissinger (2013)* deploy multivariate statistical process control for data integrity, utilizing principal component analysis to distinguish between regular and irregular power system subspaces for information leakage analysis. *Yang et al. (2020)* introduce a Conditional Deep Belief Network (CDBN) based intrusion detection mechanism to quickly and accurately identify attack features, demonstrating the model's effectiveness in real-time detection. *Sahu et al. (2021)* combine network and physical data for intrusion detection, testing the model's efficacy with various learning approaches proving its accuracy in identifying network intrusions to safeguard the power system.

However, the dynamic nature of keys may require robust mechanisms for key distribution, rotation, and storage, which can be challenging to implement and maintain. Cryptographic operations, such as message authentication code generation, can introduce computational overhead. Anomaly detection techniques, like difference sequence variance, may lead to false positives. Therefore, we develop SH-SDS for automated security detection in substations, structuring detection into two phases: initial strategy library creation *via* RG Algorithm and detailed item-by-item testing using the SG algorithm. This process yields a targeted detection list, with SH-SDS comparing outcomes against set benchmarks to generate comprehensive reports. Demonstrated practical efficiency and effectiveness underscore SH-SDS's role as a pivotal security tool for substations.

**Table 1 The supplementary insights into recent researches on IDSs include models or systems, methods used, and methods for solving problems.**

| Ref. (Year) | Model/System | Methods | Solving problems |
|---|---|---|---|
| Chen et al. (2017) | Secure operating environment measurement framework | Security baseline review and environmental vulnerability detection | Information security issues of mobile terminals |
| Zhang & Liu (2021) | Network security data detection | Immune algorithm (immune environment initialization, feature extraction, immune evolution and immune detection) | Detect the security problems in real-time network data |
| Kai, Qiang & Yixuan (2021) | Network security perception system | GA-Elman neural network | Security prediction and early warning |
| Zou et al. (2019) | Security defense system model | Intelligent substation network security defense system model | Feasibility and effectiveness of the defense system |
| Fang et al. (2013) | Adaptive control theory (NSVMAC) | Markov decision process and Q-Learning | Dynamic network environment |
| Xiang et al. (2022) | Semantic enhanced network | Integrating CNN and BiLSTM | I-S secondary system |
| Xu, Zhou & He (2022) | CL-IDS model | CL-IDS composite neural network intrusion detection model | Poor generalization |

## PROPOSED METHOD: SH-SDS

### Substation networks

Substations, comprising primary and secondary systems, are crucial in transforming, distributing, and controlling electrical energy. The primary system comprises circuit breakers, main transformers, and other electrical equipment for power transmission. The secondary system monitors, controls, and protects the primary equipment, consisting of hosts and embedded devices carrying different functions connected through information networks (Matta et al., 2012). The master station receives the status of equipment from the secondary system of the substations and issues control commands for primary and secondary equipment in the substations. The overview of the network between substations and master stations is illustrated in Fig. 2.

The described architecture of a substation provides an overview of the network configuration and security measures in place. Here are some key components for network security protection in substations.

### Cyber security threat model in substation

Power grid operation data, equipment operating permissions, and the stable operation of the power system may all be targets of attackers. Attackers may attack the power grid through network or social engineering means. Threat modeling is a technique for identifying security risks. For substation network security, threat modeling assesses all potential methods that could compromise system confidentiality, integrity, and availability from an attacker's viewpoint, thereby mitigating such threats. This article utilizes the STRIDE model for network security threat modeling, analyzing risks related to spoofing, tampering, repudiation, information disclosure, denial of service, and evaluation of privilege, as shown in Table 2 (Sheikh & Singh, 2022). Some possible protective measures are also listed.

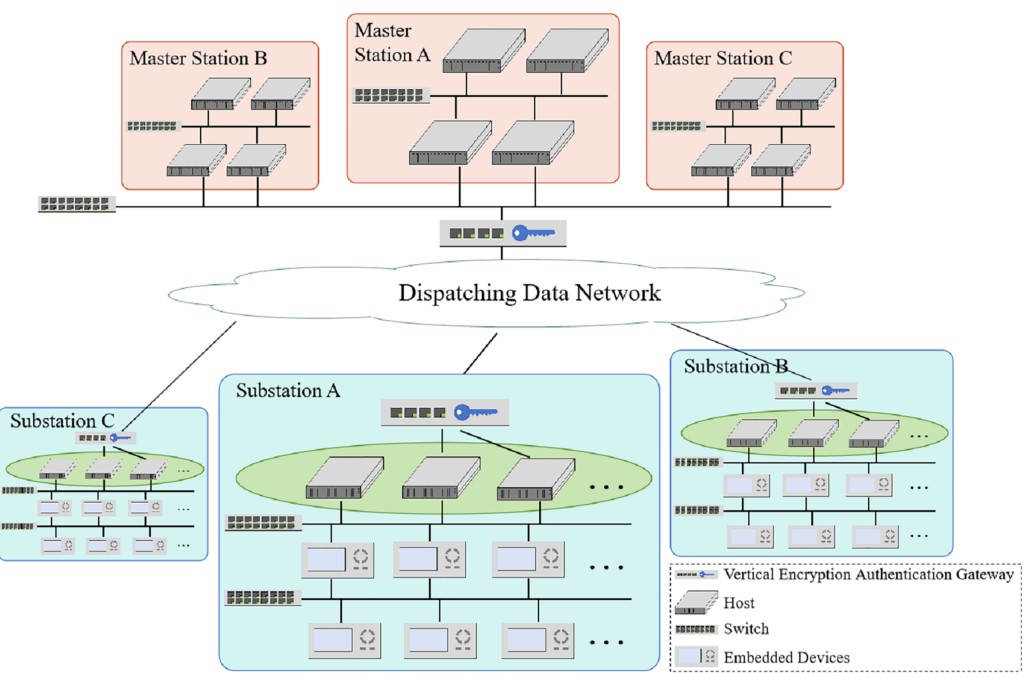

**Figure 2 The overview of a network between substations and master stations, composed of substations and master stations, is connected by dispatching a data network.**

**Table 2 The cyber security threat model in substation, including spoofing, tampering, repudiation, information disclosure, denial of service and evaluation of privilege.**

| Threat | Definition | Possible protective measures |
|---|---|---|
| Spoofing | Disguise as a legitimate user, process, or system element | Delete expired, useless, or hidden accounts. Set user privileges as required. Forbid default username. Increase password complexity. |
| Tampering | Illegal modification or editing of information | Forbid high risk ports and useless services. Logs audit. |
| Repudiation | Deny certain operations performed in the system | Logs audit. |
| Information disclosure | Data leakage or unauthorized access to confidential information | Increase password complexity. Forbid high risk ports and useless services. |
| Denial of service | Interrupt services for legitimate users | White list. Forbid high risk ports and useless services. Lock the terminal when not in use. Clear history command. |
| Evaluation of privilege | Users with restricted permissions gain higher access to system elements | Set user privileges as required. Disable super accounts. |

The STRIDE model outlines cyber threats to substations and some countermeasures. The intrusion through the dispatching data network can be effectively prevented by the dedicated vertical encryption authentication gateway, so we focus on the near-source attacks of intruders. After entering the substation, intruders can directly connect their computers to the internal network of the substation and obtain the addresses of other hosts in the substation through address sniffing, or obtain operational permissions and even administrator privileges of any host in the substation through dictionary attacks, brute

force cracking, and other methods. Furthermore, intruders can remotely connect to the host that can control the primary equipment of the substation through high-risk ports or SSH, operate the power grid equipment and cause power grid failures. Here are some key components for network security protection in substations.

- **Network configuration.** The substation architecture involves a local network connected to an external WAN (Wide Area Network) through a router, which serves as the gateway between the potentially less secure external WAN and the internal local network of the substation.
- **Perimeter security.** The router connecting the local network to the WAN is secured using conventional perimeter security measures.
- **Local network division.** The local network is subdivided into two main parts: the guard net and the automation network. The guard net is designed to contain IEDs responsible for safeguarding the energy distribution network. These devices focus on detecting and responding to electrical issues such as short circuits or ground faults. The automation network connects IEDs and RTUs (Remote Terminal Units) and facilitates automated operations within the substation. While the two parts (guard net and automation network) work independently, they are physically connected within the communication network. It implies that there is a level of connectivity between the guard net and the automation network. The physical connection between them suggests the need for careful design and security measures to prevent unauthorized access or potential security vulnerabilities.

Static detection strategies can effectively detect missing or incorrect configuration items for network security protection measures such as password complexity and audit logs, but they have limited accuracy when facing network security protection measures such as high-risk port disabling and whitelisting. Dynamic detection strategies can effectively solve this problem. However, dynamic detection strategies are also powerless when facing static problems. Therefore, a static-dynamic strategy is necessary.

## Substation host security detection method structure

The architecture of our SH-SDS is illustrated in Fig. 3. SH-SDS consists of five modules: (1) Data generation module, (2) strategy generation module, (3) detection module, (4) report module, and (5) user interaction module.

These modules are briefly described as follows.

1. **Data generation module.** We collect all substations' security detection items, requirements, and methods. We build an index library to store the substation security detection. In the future, if there are new security detection items, they can be added directly to the library for storage. At the same time, new items can be obtained from the library for substation host security detection without reconfiguration.
2. **Strategy generation module.** We employ Algorithm 2 with the target substation host to effectively generate a static-dynamic strategy from the substation host security detection

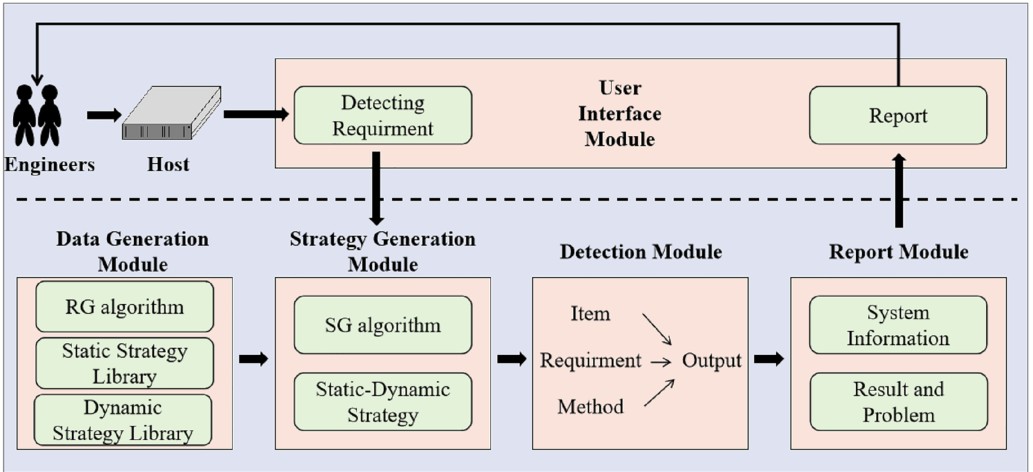

**Figure 3 The proposed architecture of SH-SDS comprises five modules of data generation module, strategy generation module, detection module, report module and user interface module, input detecting requirement and output detection report.**

library. Static-dynamic strategy for target substation host is used to conduct security detection, which can conduct comprehensive testing.

3. **Detection module.** SH-SDS performs target substation host security detection based on the generated static-dynamic strategy. For each item to be tested, we use the specified method to check item by item whether it meets safety requirements.

4. **Report module.** After the detection of all items, SH-SDS reports the result, which includes system information, IP, and analysis.

5. **User interaction module.** SH-SDS displays the graphic interface for engineers. Engineers can start detection using the graphic interface. Users can also get a report of detection through the user interaction module. The user interaction module allows simpler operations.

These modules are very significant for the security detection of hosts in substations. The detection is conducted by SH-SDS instead of engineers, improving the speed and accuracy of security detection.

## Data collection and preprocessing

### Standard library data generation

The substation host system has extremely high requirements for network security. China has formulated a series of relevant network security level protection standard documents like DL/T 2613-2023 and DL/T 2614-2023 for the network security requirements of host system (*Haixiang, Hongyang & Wenming, 2023*; *Haixiang, Wenming & Hongyang, 2023*). The data used in this article comes from the section on host system security detection in the standard documents of classified protection for cybersecurity. For example, in identity verification, there are complexity requirements, such as the password length being no less than eight characters and presenting as a mixture of letters, numbers, or special characters. The identity verification information should be replaced regularly. In addition, there are

requirements for the number of illegal logins, and they should be automatically pushed out when the login timeout occurs. In terms of security auditing, it is required to enable auditing functions, record information such as dates, times, and users of events, and ensure that the audit retention time meets legal and regulatory requirements. In terms of intrusion prevention, it is required to shut down unnecessary system services, default shares, and high-risk ports (*Jianyong, Hongsheng & Baoxi, 2018*).

The following collection process for the standard library involves a series of streamlined steps, transforming the unstructured data into a structured format for further use.

- Find each six-level title in standard documents of classified protection for cybersecurity (such as 6.1.3.2.3.1), and divide the document into multiple fragments according to the six-level title.
- In the fragment, retrieve the content between 'evaluation object' and 'evaluation implementation'. If the 'terminal' or 'server' field does not appear, delete the entire fragment, and if there is, retain the entire fragment and record it as F, as shown in Eq. (1).

$$F = \{f_1, f_2, ..., f_n\} \tag{1}$$

- Segment each $f_n$ in F. As the first step in natural language processing, sentences composed of character sequences are recombined into a set of words through word segmentation. Chinese word segmentation is dividing a Chinese sentence into a set of words. We apply the Jieba word segmentation library (https://github.com/fxsjy/jieba) and processed $F = \{f_1, f_2, ..., f_n\}$ as *SE*, as shown in Eq. (2).

$$SE = \{se_1, se_2, ..., se_n\}. \tag{2}$$

- Remove the stop words for each $se_n$ in *SE*. Many punctuation marks, modal particles and other words in the text have no practical effect on understanding. These should be removed from the word segmentation results, which can save storage space, reduce the noise caused by understanding sentences, and reduce the dimension of the text. We apply the hit_stopwords list (https://github.com/goto456/stopwords.git) to replace the stop words in *SE* with spaces, and get *S* as Eq. (3).

$$S = \{s_1, s_2, ..., s_n\}. \tag{3}$$

- Extract the content between 'evaluation object' and 'evaluation implementation' in $s_n$ as $item_n$.
- Extract the content between 'evaluation implementation' and 'unit determination' in $s_n$ as $content_n$.
- Combine $item_n$ and $content_n$ as $DI_n$, as shown in Eq. (4).

$$DI_n = (item_n, content_n). \tag{4}$$

- Put all the $DI_n$ together to form a data set *DI*, as shown in Eq. (5).

$$DI = \{DI_1, DI_2, ..., DI_n\} = \{(item_1, content_1), (item_2, content_2), ..., (item_n, content_n)\}. \quad (5)$$

Subsequently, we develop detection methods based on the configuration document of the target host system and classify them into static detection and dynamic detection, which are used to construct a static strategy library and a dynamic strategy library. Finally, based on these libraries, the detection strategy is generated using SG algorithms. Therefore, we have two following libraries, the data of the (i) static strategy library, and (ii) dynamic strategy library.

Strategies in the static strategy library are the host network security detection methods for checking configurations, while strategies in the dynamic strategy library are host network security detection methods that simulate real attacks. The details are described as follows.

### Static strategy library data generation

Static strategy, also known as passive detection technology, refers to the host network security detection method that checks the configuration of network security protection parameters, mainly including text matching and strategy output inspection (*Kim et al., 2019*). Different detection methods exist for static substation host security detection for different types of items. In terms of encryption, it mainly stipulates the password length, the password characters' composition, and the update cycle according to the requirements.

We generate requirements using the RG algorithm (Algorithm 1), which is used to construct the strategy library. The algorithm of construction is shown below.

The process of building a static strategy library consists of four steps.

1) Find host strategies that need to be tested by checking the configuration file as target items $TI$, and stored them as $SI$ in terms of static strategy library.
2) Get requirements $RI$ and detection methods $MI$ by using Algorithm 1.
3) Store $RI$ and $MI$ we get in step (2) as $SCR$ and $SDM$ respectively.
4) Construct the static strategy library $SL = \{SL_1, SL_2, SL_3, ..., SL_n\}$, where $SL_k = (SI_k, SCR_k, SDM_k)$, $SI_k$ is the k-*th* item in the static library, $SCR_k$ is the configuration requirement of $SI_k$, $SDM_k$ is the detection method for $SI_k$. A few examples are presented in Table 3.

### Dynamic strategy library data generation

Dynamic strategy, also known as active detection technology, refers to the host network security detection method that simulates the actual attack behavior, mainly including port tests, service tests, and so on (*Chen et al., 2014*). There are mainly real-time vulnerability detection technology and risk analysis technology for the dynamic detection strategy of substation hosts. As for access control, measures such as closing services and ports are usually used. We construct a dynamic strategy library for automatic strategy generation. The process of building a dynamic library involves the following steps.

1) Find host strategies that need to be tested by checking the configuration file as target items $TI$, and stored them as $DI$ in terms of dynamic strategy library.

| **Algorithm 1** RG algorithm for the generation of strategy library data. |
|---|

**INPUT:** List of target items $TI = \{TI_1, TI_2, TI_3, \ldots, TI_n\}$, Relevant standard documents of classified protection for cybersecurity $DI = \{DI_1, DI_2, DI_3, \ldots, DI_n\}$

**OUTPUT:** Requirements RI=$\{RI_1, RI_2, RI_3, \ldots, RI_n\}$, Methods MI=$\{MI_1, MI_2, MI_3, \ldots, MI_n\}$

1. **For** i = 1:n **do**

2.    **For** j = 1:n **do**

3.       /*Confirm corresponding requirements and detection methods*/

4.       **If** $TI_i$ matches items in $DI_j$ **then**

5.          Append requirements in $DI_j$ to $RI_{ij}$ and append methods in $DI_j$ to $MI_{ij}$

7.       **Else** Append $TI_i$ to NoMatch

8.       **End if**

9.    Combine $RI_{i1}$ to $RI_{ij}$ as $RI_i$ and combine $MI_{i1}$ to $MRI_{ij}$ as $MI_i$

10.    **End for**

11. **End for**

**Table 3 Static strategy library *SL* (example).**

| k (ID) | $SI_k$ (Target items) | $SCR_k$ (Requirements) | $SDM_k$ (Detection methods) |
|---|---|---|---|
| 1 | Complexity of system login password | Mixed combination of letters, numbers and special characters | Matching by 'dcredit', 'ucredit', 'lcredit' and 'ocredit' fields in the '/etc/pam.d' file |
| 2 | Update cycle of system login password | Less than 30 days | Matching by 'PASS_MAX_DAYS' field in the '/etc/pam.d' file |
| 3 | The timeout lock time after system login | Less than 10 min | Matching by 'TMOUT' field in the '/etc/profile' file |
| 4 | SSH log audit | Run the SSH log audit function | Matching by 'SyslogFacility AUTH' field in the '/etc/ssh/sshd_config' file |
| 5 | The retention time of audit content | More than 6 months | Matching by 'rotate' field in the '/etc/logrotate.conf' file |
| … | … | … | … |

2) Get requirements *RI* and detection methods *MI* by using Algorithm 1.

3) Store *RI* and *MI* we get in step (2) as *DCR* and *DDM* respectively.

4) Construct the dynamic strategy library $DL = \{DL_1, DL_2, DL_3, \ldots, DL_n\}$, where $DL_k = (DI_k, DCR_k, DDM_k)$, $DI_k$ is the $k$-th item in the dynamic library, $DCR_k$ is the configuration requirement of $DI_k$, $DDM_k$ is the detection method for $DI_k$. A few examples are presented in Table 4.

## Strategy generalization

The traditional host network security detection strategy is mainly focused on static strategies. However, only static strategies are insufficient; some items must be detected through dynamic strategies. Thus, building an active detection strategy that combines

**Table 4 Dynamic strategy library *DL* (example).**

| k (ID) | $DI_k$ (Target items) | $DCR_k$ (Requirements) | $DDM_k$ (Detection methods) |
|---|---|---|---|
| 1 | Opening ports | Closing high-risk ports such as 135 | Scanning ports of the target IP |
| 2 | White list | Set whitelists to limit access to unfamiliar clients | Simulate non-whitelist client access |
| … | … | … | … |

dynamic and static strategies is necessary, making the substation host security detection more comprehensive.

When a substation host security inspection work is carried out, we need to obtain a target host security inspection project file first. Then, the items in the file are matched with the static strategy library and the dynamic strategy library to obtain the security detection configuration file, which is the target host's security detection strategy. The strategy generation algorithm we use is shown in Algorithm 2.

Algorithm 2 is stable and the time complexity is O(n), where n is the length of target items *TI*.

In Algorithm 2, we mention matching, which is used to find if the target item is in the library. When we need to match the target item with the item in the library, firstly, we set the target items as *TI* and the items in the library as *LI*. For computers, items are strings, actually. Thus, in the process of computers, *TI* is stored as Eq. (6) and *LI* is stored as Eq. (7).

$$TI = TI_0 TI_1 \ldots TI_{n-1} \tag{6}$$
$$LI = LI_0 LI_1 \ldots LI_{n-1} \tag{7}$$

The first step we need to know if $TI_m$ matches $LI_n$ is to find the prefix and suffix with the most extended length in *TI*. If there is $TI_0 TI_1 \ldots TI_{k-1} TI_k = TI_{j-k} TI_{j-k-1} \ldots TI_{j-1} TI_j$, there are identical prefixes and suffixes with a maximum length of $k + 1$. Then we need to construct an array named $\alpha$. $\alpha$ is as Eq. (8).

$$\alpha = \begin{cases} 0 \text{ when } j = 1 \\ 1 \text{ unmatch} \\ max\{k|1 < k < j \text{ and } TI_0 TI_1 \ldots TI_{k-1} TI_k = TI_{j-k} TI_{j-k-1} \ldots TI_{j-1} TI_j\} \end{cases} \tag{8}$$

When *TI* unmatched *LI*, j = $\alpha$[j], the number of *TI* moves to the right is j − $\alpha$[j] = 0. When $TI_{j-k} TI_{j-k-1} \ldots TI_{j-1} TI_j$ matched $LI_{i-k} LI_{i-k-1} \ldots LI_{i-1} li_i$ as well as $TI_j$ unmatched $LI_i$, $\alpha$[j] = k, which means the same prefix and suffix with $TI_j$ maximum length of k in $LI_i$ that does not contain $TI_j$. That means as Eq. (9).

$$TI_0 TI_1 \ldots TI_{k-1} TI_k = TI_{j-k} TI_{j-k-1} \ldots TI_{j-1} TI_j \tag{9}$$

We set the j = $\alpha$[j], and let *TI* moves j − $\alpha$[j] to the right, which $TI_0 TI_1 \ldots TI_{k-1}$ matches $LI_0 LI_1 \ldots LI_{j-1}$. Then let $TI_k$ continue to match $LI_j$.

---

**Algorithm 2** SG algorithm for the generation of detection strategy.

**INPUT:** List of target items TI={$TI_1$, $TI_2$, $TI_3$,…, $TI_n$}, static strategy library SL={$SL_1$, $SL_2$, $SL_3$,…, $SL_n$}, dynamic strategy library DL={$DL_1$, $DL_2$, $DL_3$,…, $DL_n$};

**OUTPUT:** Security detection strategy for the target host SDS = {$SDS_1$, $SDS_2$, $SDS_3$,…, $SDS_n$}, where $SDS_k$= ($TI_k$, $CR_k$, $DM_k$), $TI_k$ is the kth target item, $CR_k$ is the configuration requirement of $TI_k$ and $DM_k$ is the detection method for $TI_k$;

1. **For** i = 1:n **do**

2.     **For** j = 1:(length of SL) **do**

3.         **If** ($TI_i$ matches $SI_j$) **then**

4.             Append $SL_j$= ($SI_j$, $SCR_j$, $SDM_j$) to SDS

5.         **Else if** (no match for $TI_i$) **then**

6.             Append $TI_i$ to NoMatch

7.     **End if**, **End for**

8.     **For** k = 1:(length of DL) **do**

9.         **If** ($TI_i$ matches $DI_k$) **then**

10.             Append $DL_k$= ($DI_k$, $DCR_k$, $DDM_k$) to SDS

11.         **Else if** (no match for $TI_i$) **then**

12.             Append $TI_i$ to NoMatch

13. **End if**, **End for**, **End for**

14. **For** item in NoMatch /* add unknown items into our library */

15.     Discriminate whether to use static or dynamic methods for this item

16.     Set the configuration requirement as $CR_{item}$ of item manually

17.     Set the detection method as $DM_{item}$ for item manually

18.     Append (item, $CR_{item}$, $DM_{item}$) to SDS

19.     **If** static method **then**

20.         Append (item, $CR_{item}$, $DM_{item}$) to SL

21.     **End if**

22.     **If** dynamic method **then**

23.         Append (item, $CR_{item}$, $DM_{item}$) to DL

24. **End if**, **End for**

---

The complexity of this calculation process is O(m + n), where m is the length of *TI* and n is the length of *LI*. The best complexity performs as O(m) while the worst complexity performs as O($m^2$). We use this calculation method to find the item that matches the target items in the static and dynamic libraries. Then, we use Algorithm 2 to generate a static and dynamic detection strategy. The security detection configuration file is obtained by matching the host network security detection project file with the static and dynamic strategy libraries. It serves as a strategy file used in security detection.

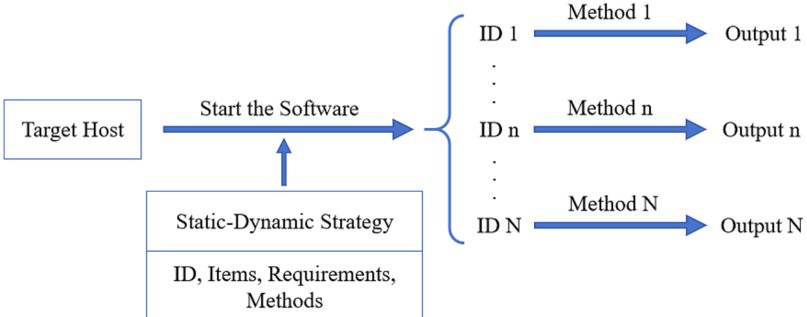

**Figure 4 The process of security detection.** Initially, the software was executed to detect the target host using a static-dynamic strategy. Then, the result of security detection of the target host is output to the user interface.

## Security detection

According to the generated dynamic and static detection strategy, the security detection of the target host of the substation is carried out. The flow chart is shown in Fig. 4.

Firstly, we must configure the target host's dynamic and static strategy files and start SH-SDS. Secondly, the target items and their ID, configuration requirements, and detection methods are identified automatically. Finally, the configuration requirements of each item are compared according to the detection method in the strategy file, and the detection results are obtained.

The inspection report is summarized based on the results of each project obtained by network safety inspection. The report's contents include system information, device IP, results, and problems with various test items. Through the security detection report, users can notice the security detection results for the target host and correct the returned problems. The whole process of safety detection is efficient and automatic.

## EXPERIMENTS AND RESULTS

The experimental workflow of the SH-SDS, aligning with our substation host security detection and detailed in the results section, is concisely outlined in the subsequent steps, which is depicted in Fig. 5.

The following are the critical parts of our experimental implementation.

- **Data collection.** Gather standard documents, including national and industry standards.
- **Preprocessing.** Extract the requirements for the substation host from the standard file as detection items.
- **Library data generation.** Refine the detection features based on the preprocessed data and apply Algorithm 1 to generate static and dynamic strategy library data.
- **Detection model.** Generate detection models binding hosts' strategy configuration data and Algorithm 2 based on static-dynamic strategy.
- **Security detection.** Apply a detection model to perform security detection on hosts.
- **Performance evaluation.** Evaluate the method using metrics such as software size, CPU utilization, response time, accuracy, *etc.*

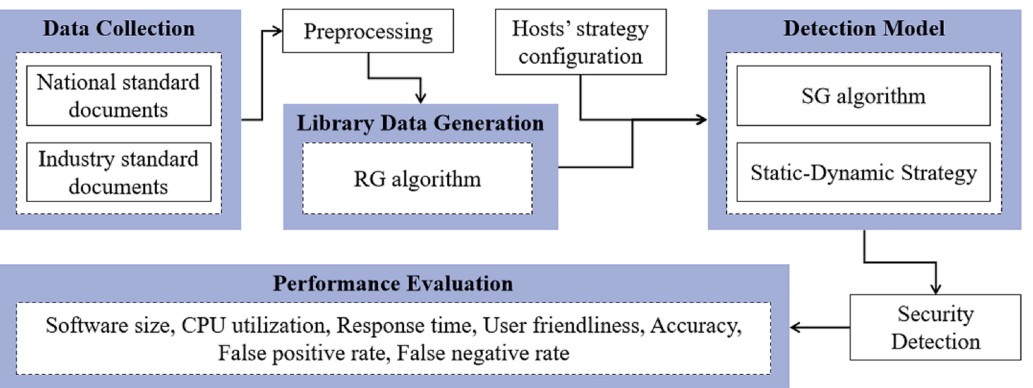

**Figure 5** Experimental flowchart of the proposed SH-SDS method, including data collection, library data generation, detection model, security detection, and performance evaluation.

## System performance testing

We conduct performance testing and quantitative evaluation of SH-SDS. We conduct tests in both simulated and real environments. In simulated environment, we conduct the performance testing configuration, the CPU is i7-10610U, the memory is 4GB, and the operating system includes Ubuntu 16.04.7, Ubuntu 22.04.1, and Kylin 3.3. In real environment, the CPU is Hygon 3285, the memory is 64GB and the operating system is Kylin 3.3. We conduct multiple experiments in simulated environment, including the SH-SDS, the openSCAP software, and manual inspection. The openSCAP is an open-source host automatic scanning tool developed by Redhat, mainly composed of tools and baseline libraries. It maintains great flexibility and interoperability, making it widely used in Linux systems (*Liu & Hu, 2023*). After configuring detection content in the baseline library of openSCAP, we perform security checks on the host through the terminal. Manual inspection is based on the requirements of the detection items, manually entering commands one by one in the terminal according to the prescribed method and manually judging whether the host is safe based on the returned results.

We conduct statistics and analysis on the results of the same host security testing items. The calculation methods for various items are as follows. The software size refers to the amount of space occupied by the software on the system disk, which is composed of code segment size and data segment size. It can be viewed using the 'ls – l' command. The CPU utilization is shown in Eq. (10) (*Yu-Wei, Dong & Shu-Cheng, 2018*). The response time is the time it takes for the user to initiate a request and receive the result returned from the software. The user-friendliness is evaluated based on whether the software has an interface and whether the interface is convenient for human–computer interaction. The accuracy is calculated as Eq. (11). The false positive rate is calculated as Eq. (12) while the false negative rate is calculated as Eq. (13). We obtain the following performance test results, as shown in Table 5.

$$CPU\ utilization = \frac{the\ usage\ time\ of\ CPU}{total\ running\ time\ of\ the\ software} \times 100\% \tag{10}$$

**Table 5 Performance testing results of software size, CPU utilization, response time, user-friendliness, and accuracy among SH-SDS, OpenSCAP, and manual inspection.**

| Software evaluation | SH-SDS | OpenSCAP | Manual inspection |
|---|---|---|---|
| Software size | 9.9 MB | 27.13 MB | / |
| CPU utilization | 2.0% | 75.8% | / |
| Response time | 0.144s | 0.691s | 2,100s |
| User friendliness | √ | × | × |
| Accuracy | 99.735% | 99.470% | 98.497% |

$$Accuracy = \frac{\sum_{i=1}^{n} \frac{number\ of\ items\ with\ correct\ results\ in\ test\ i}{total\ number\ of\ items\ in\ i}}{n} \times 100\% \tag{11}$$

$$False\ positive\ rate\ (FPR) = \frac{FP}{FP\ +\ TN} \times 100\% \tag{12}$$

$$False\ negative\ rate\ (FNR) = \frac{FN}{FN\ +\ TP} \times 100\%. \tag{13}$$

From the test results, we can see that both the size of SH-SDS and CPU utilization are smaller than that of openSCAP. Regarding response time, SH-SDS is shorter than openSCAP and takes much less time than manual inspection. Regarding user-friendliness, SH-SDS has developed a graphical interface to display and export detection results. At the same time, openSCAP software does not have a graphical interface, which is not intuitive regarding results display. In terms of accuracy, SH-SDS and openSCAP perform relatively well, both of which are better than the accuracy of manual inspection.

## Performance and accuracy

In a simulated environment, we test the speed and accuracy of different security detection items, including the complexity of system login password, update cycle of system login password, and other 12 items, which are the most important in substation host security detection. We construct dynamic and static combination test projects in the test data set and adjusted the number of test items. The test results are depicted in Fig. 6.

In the simulated tests, the SH-SDS demonstrated remarkable consistency in response time across varying numbers of detection items, as depicted in Fig. 6A. The blue background represents the confidence interval around the mean response time, indicating the variability and reliability of the measurements across different numbers of items tested. This uniformity in performance indicates the software's stability in operation. Furthermore, Fig. 6B reveals that the accuracy of the detection tests is exceptionally high, nearing 100% across all trials. These findings underscore the robustness of SH-SDS, which maintains high accuracy without compromising speed, even as the complexity of the tasks varies. This balance of speed and precision showcases the system's efficacy in dynamic and static security detection scenarios.

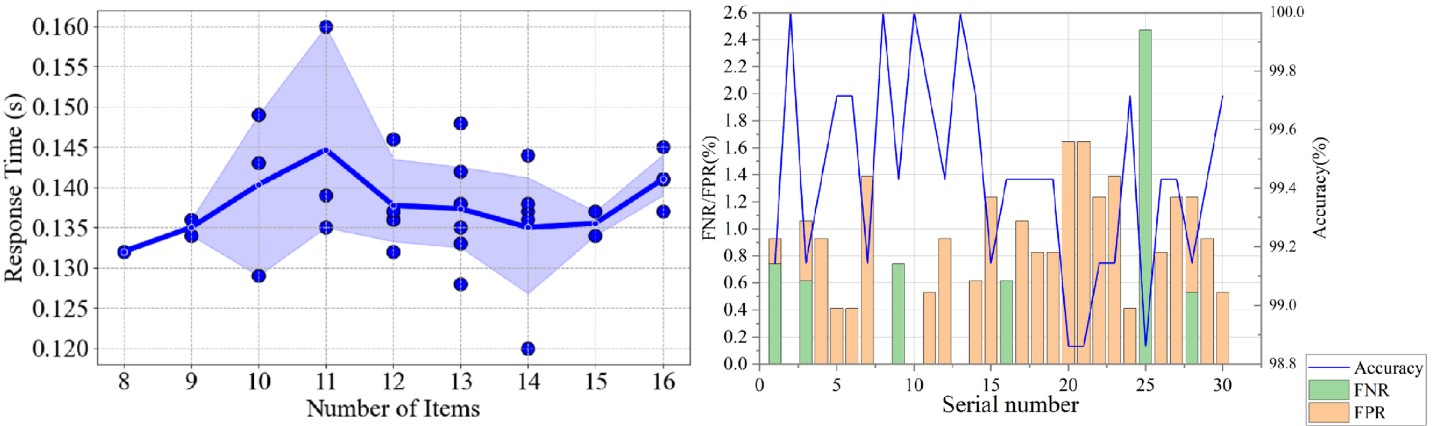

**Figure 6** The performance of (A) response time(s) and (B) accuracy (%), FPR (%) and FNR (%) of proposed SH-SDS work running in a simulated environment with different numbers of items.

**Table 6 Comparisons of ablation experiments using static, dynamic, and our dynamic-static strategies.**

| Strategy | Metrics | Only static | Only dynamic | Random |
|---|---|---|---|---|
| Static strategy | Accuracy | 99.67% | 5.13% | 89.74% |
| | Response time | 0.121s | 0.123s | 0.127s |
| Dynamic strategy | Accuracy | 1.28% | 97.43% | 11.39% |
| | Response time | 0.02s | 0.019s | 0.02s |
| Static-dynamic-strategy | Accuracy | 99.67% | 98.71% | 99.71% |
| | Response time | 0.133s | 0.130s | 0.129s |

We conduct ablation experiments on the above test data using static strategies, dynamic strategies, and dynamic-static strategies. The experimental results are shown in Table 6.

Table 6 reveals that the static strategy excels with purely static items, reaching 99.67% accuracy. However, it falters to 5.13% with only dynamic items. Conversely, the dynamic strategy soars to 97.43% accuracy with dynamic items but drops significantly with static ones. The integrated static-dynamic strategy outperforms both, demonstrating superior adaptability and robustness across varied test conditions, indicating its comprehensive efficacy in security detection.

In addition, we also test it in a real environment and compare it with the traditional manual detection method. The test results are illustrated in Fig. 7.

In Fig. 7A, the proposed SH-SDS exhibits the highest accuracy at Substation C. It surpasses manual detection at all sites, with the most significant improvement at Substation C, which is nearly 100%. This enhanced accuracy stems from SH-SDS's automation and sophisticated algorithms, which mitigate human error and deliver consistent, high-precision security checks across substations. Based on detection time Fig. 7B, the proposed SH-SDS has the lowest detection time at Substation A. It
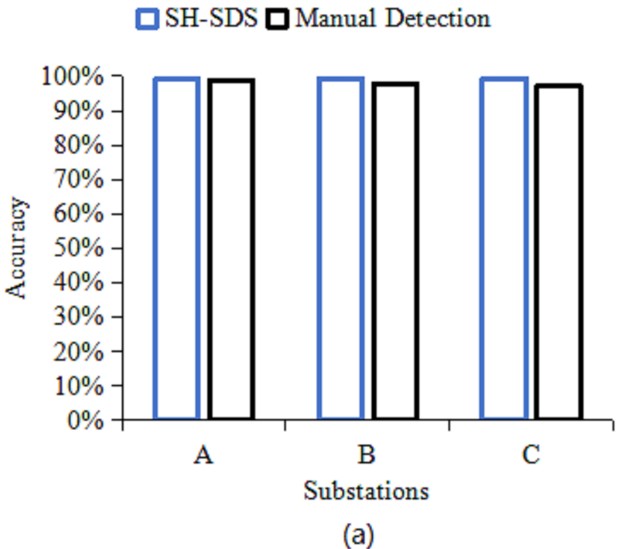
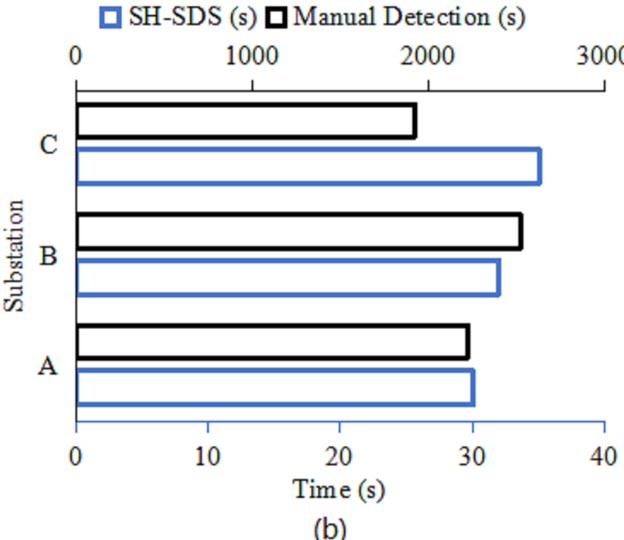

**Figure 7** Comparisons of (A) accuracy (%) and (B) detection time(s) performance testing in a real environment.

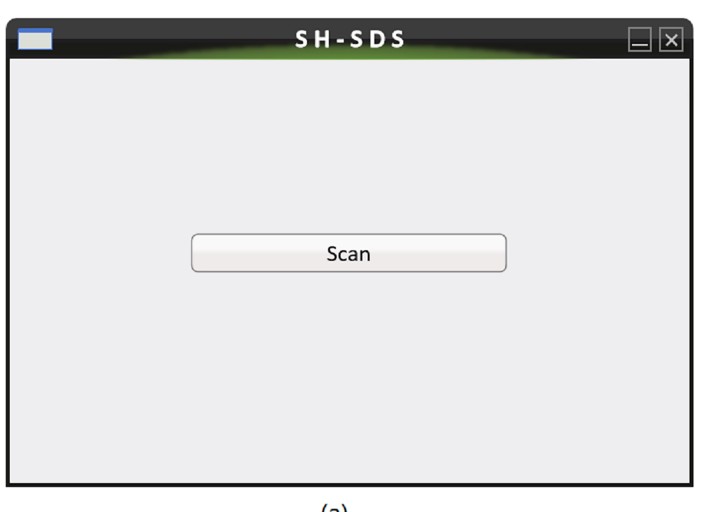
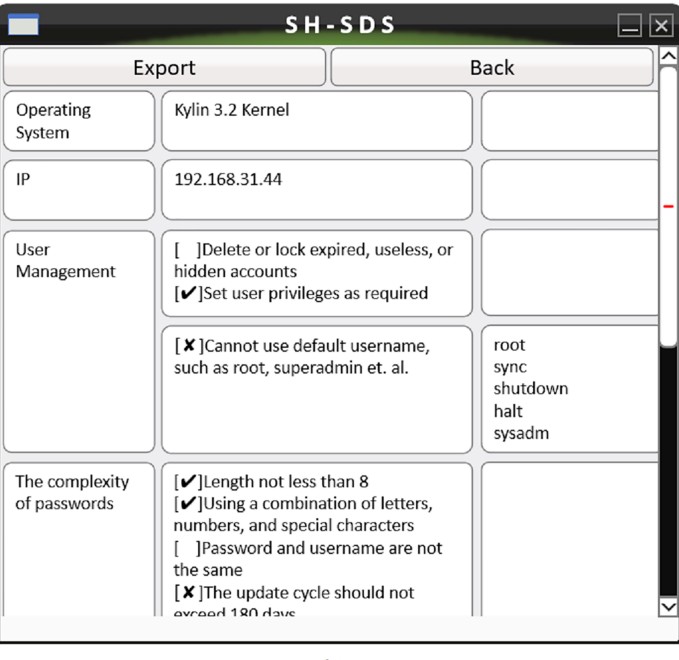

**Figure 8** The system's main interface (A) is used to initiate inspection. The system operation interface (B) is used to view and export inspection results.

outperforms manual detection at all substations, most notably at Substation C, which is approximately 98% faster. This significant efficiency gain is due to the automation of the detection process, which minimizes human intervention and speeds up the overall time required for security checks, showing the proposed SH-SDS's capacity for rapid analysis and response in substation security.

### System operability test

We use an actual host security detection operation to show the software's operation process, as follows.

1) After copying the software to the host, we run the software using the command terminal.
2) After we open the software, click the 'scan' button to detect the host, as shown in Fig. 8A.
3) After the scanning is completed, we can view the inspection results on the software interface, or output the scanning results in the specified format, as shown in Fig. 8B.
4) After completing the rectification of the abnormal items scanned out, we click 'back' button to return to the main interface and the detection can be performed again.

## CONCLUSION

The substation host controls and protects power system equipment, and improving informatization level of substation host also brings higher network security risks. Therefore, detecting the network security of the substation host is very important. In this article, we implement SH-SDS for substation host security detection. After practical use, the network security detection of the substation host has been improved in speed and accuracy.

SH-SDS can also be used for LAN office computer network security testing. However, the software is not easy to maintain. In the face of new network security detection requirements, it is necessary to update the network security detection standard library manually. Moreover, the software is imperfect in the human factor engineering of interface interaction, and the software operation is not convenient enough. In the future, we will collect more host system network security detection projects to enrich our substation host system network security detection standard library. At the same time, we will also study how to automatically extract the latest network security detection requirements from related standards, cases, reports, *etc.*, and automatically update the network security detection standard library. In addition, the software's interface also needs to be enhanced.

### Funding

This work was supported by Shenzhen Polytechnic Research Fund (Nos. 6023310010K and 6023310009K). The funders had no role in study design, data collection and analysis, decision to publish, or preparation of the manuscript.

## Grant Disclosures

The following grant information was disclosed by the authors:
Shenzhen Polytechnic Research Fund: 6023310010K, 6023310009K.

## Competing Interests

Yang Diao and Wei Liu are employed by Shaoguan Power Supply Bureau, Guangdong
Power Grid Co., Ltd.

## Author Contributions

- Yang Diao conceived and designed the experiments, performed the experiments,
  analyzed the data, performed the computation work, prepared figures and/or tables,
  authored or reviewed drafts of the article, and approved the final draft.
- Hui Chen conceived and designed the experiments, analyzed the data, authored or
  reviewed drafts of the article, and approved the final draft.
- Wei Liu conceived and designed the experiments, performed the computation work,
  authored or reviewed drafts of the article, and approved the final draft.
- Abdur Rasool performed the computation work, prepared figures and/or tables, and
  approved the final draft.

## Data Availability

   The SH-SDS software along with its source code and instruction are available at
Zenodo: Diao, Y. (2024). SH-SDS. Zenodo. https://doi.org/10.5281/zenodo.13919924.
   The raw data is available in the Supplemental Files.

## Supplemental Information

Supplemental information for this article can be found online at http://dx.doi.org/10.7717/
peerj-cs.2512#supplemental-information.

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
