# Peer review of "SH-SDS: a new static-dynamic strategy for substation host security detection"

_PeerJ Computer Science, doi:10.7717/peerj-cs.2512_

## Round 0.1 · original submission · Major Revisions

Please address the comments from the reviewers, especially the first reviewer when preparing your revised manuscript.

Reviewer 1 ·

Basic reporting

This paper studies a new methodology for enhancing security detection in power substations. Overall, this paper is well-organized, and the writing quality is good.

This paper provides a comprehensive literature review of smart substation architectures and cyberattack detection systems.

I like the figures in this paper, as they are clear to read and follow for me. However, I still have few format suggestions to improve the paper’s presentation and readability:
1) Add borders/lines for each row in Table 1 and 2;
2) Add the color theme for the code listing in Algorithm 1 and 2;

Experimental design

I think this paper is clear in filling the identified research gap in network security for substations. However, one unclear point is the lack of a concrete threat model. Since the literature review covers a wide range of attacks and defenses, I suggest authors to clearly define the threat model at least including attacker’s goal, knowledge, capability and prerequisites.

There are some important details missing in the experiments. First, the authors claim that they conducted tests in both simulated and real environments. However, the system details for the simulator and real-world environments are not provided. Second, Tables 4 and 5 show the detection results, but the details of the attack cases themselves are missing. It is unclear how these attacks were implemented and how they functioned.

This paper needs to consider more metrics in the experimental results. The current focus on a single metric, "accuracy," is not sufficient to demonstrate the effectiveness of the security detector. I expect to see additional metrics such as false positive and false negative rates.

Validity of the findings

The data generation processes described in lines 258 to 279 appear to be very domain specific. I am concerned that these specific string search procedures may limit the generalizability of the SH-SDS approach across different platforms. Additionally, the standard documents cited as Haixiang et al. (2023b) seem to be invalid, as I could not find any publications under that reference.

In experiments, this paper compares SH-SDS with OpenSCAP and manual inspection to validate its efficiency and effectiveness. However, I am curious why the authors chose OpenSCAP as the baseline for comparison. This is not discussed in the literature review, nor is OpenSCAP properly cited in the paper.

Authors provide the raw data file and source code with clear format.

Additional comments

No comment

Cite this review as

Reviewer 2 ·

Basic reporting

The authors don't write in parallel when they describe the bulleted lists. The paper starts out vague in the abstract and introduction before going into details. The overall structure is good.

Experimental design

The design is clearly explained, and diagrams are used well to explain how the SH-SDS works. However, the explanation of Static versus Dynamic and making a case of static-dynamic could be clarified; for example, Table 5 is more comprehensive but is presented later than in the beginning.

Validity of the findings

The results are presented in a way that shows the method/algorithm works, but the data files themselves look manually generated. How would this scale?
While this shows the data for a substation with 14 data points, what are the real-life implications for such an experiment?

Cite this review as

---

## Round 0.2 · Minor Revisions

Please address the reviewers concerns. Then the paper can be accepted.

Reviewer 1 ·

Basic reporting

To further clarify my format suggestions (i.e., color theme) in Algorithm 1 and 2, I suggest the authors to bold the necessary components based on programing language mode. For instance, we can bold “For” and “do” in the line 1 “For i = 1:n do”, and we also need to bold “End if”, “End for”, etc. This could make the algorithm listing read clearer.

Experimental design

Based on the file ‘tracked-changes-version(SH-SDS).pdf’, on page 6, line 206 – 214, the threat model needs further improvements. Threat model is not equivalent to “threat modeling”. I appreciate you list the attack targets in the revised version. However, you still need to add what exact techniques the attacker can use the perform the attacks your aim to address. You can find an example by reading the paragraph starting with “The threat model this paper focuses on, targets the integrity of critical devices” in this reference paper you already cited.

Sahu, A., Mao, Z., Wlazlo, P., Huang, H., Davis, K., Goulart, A., and Zonouz, S. (2021). Multi-source multi-domain data fusion for cyberattack detection in power systems. IEEE Access, 9:119118–119138.

Validity of the findings

Thanks for adding the FPR and FNR in Figure 6(b). However, the results of FPR and FNR look weird and problematic to me, since they are shown to be 30+% or even 100%. For instance, 100% FPR means your system cannot correctly identify the negative cases as positive.

Cite this review as

---

## Round 0.3 · accepted · Accept

You have NOT fixed the link to figure 1, which you were asked to do for the reviews. You replaced it with another link but the link is still INCORRECT. Please fix this!